# Non-Contact Monitoring of ECG in the Home Environment—Selecting Optimal Electrode Configuration

**DOI:** 10.3390/s22239475

**Published:** 2022-12-04

**Authors:** Adam Bujnowski, Kamil Osiński, Piotr Przystup, Jerzy Wtorek

**Affiliations:** 1Biomedical Engineering Department, Faculty of Electronics Telecommunication and Informatics, Gdansk University of Technology, Narutowicza 11/12, 80-233 Gdansk, Poland; 2Dynamic Precision, ul. Trzy Lipy 3, 80-172 Gdansk, Poland

**Keywords:** capacitive ECG, optimal lead configuration, motion artifacts, forward problem

## Abstract

Capacitive electrocardiography (cECG) is most often used in wearable or embedded measurement systems. The latter is considered in the paper. An optimal electrocardiographic lead, as an individual feature, was determined based on model studies. It was defined as the possibly highest value of the R-wave amplitude measured on the back of the examined person. The lead configuration was also analyzed in terms of minimizing its susceptibility to creating motion artifacts. It was found that the direction of the optimal lead coincides with the electrical axis of the heart. Moreover, the electrodes should be placed in the areas preserving the greatest voltage and at the same time characterized by the lowest gradient of the potential. Experimental studies were conducted using the developed measurement system on a group of 14 people. The ratio of the R-wave amplitude (as measured on the back and chest, using optimal leads) was less than 1 while the SNR reached at least 20 dB. These parameters allowed for high-quality QRS complex detection with a PPV of 97%. For the “worst” configurations of the leads, the signals measured were practically uninterpretable.

## 1. Introduction

Non-contact heart rate monitoring can be performed in many ways [1,2,3]. However, two techniques are the most widespread, i.e., the imaging and capacitive ECG methods. The former is based on the detection of changes in skin color caused by the variable blood supply to the tissue. Thus, it is synchronized with the heart activity. The signal can be acquired from images of the face or its fragments [1,4]. The classical approach, with the electrodes applied to the body of the examined person, is widely known as an electrocardiogram. With a non-contact approach, the measurement is performed with the use of electrodes that are not in direct contact with the body of the examined person and are embedded in furniture, e.g., a chair, armchair, bed, bathtub, car seat, or as wearable devices [5,6,7,8,9]. In this case, the electrodes are often described as capacitive. Each mentioned application demands a specific construction of the measurement system [10], e.g., to make the measurement system wearable, the electrodes are typically parts of the clothing. In general, the capacitive electrode for bioelectric potential measurements could be made of a multilayer printed circuit board [11,12] or a few layers of a conductive fabric [13]. The former solution makes it possible to locate the electronic measuring circuit (at least its input part) directly next to the capacitive electrode. Importantly, advanced shielding and processing techniques can also be used [14,15,16,17]. The latter solution takes into account the fact of the body’s changing shape at the surface and minimizes the problems of maintaining the value of coupling capacitance as high as possible. The electrode is usually separated from the body surface by some insulating material, e.g., clothes, and also very often by an air gap. Thus, the electrode, air gap, and insulating material together with the body (having relatively high electrical conductivity) form a series of connections between two capacitors. It could be accepted that the capacitance formed by the air gap is the smallest, and as a result, its value determines the total capacitance [14]. Taking into account the small electrode area, the resulting coupling capacitance is very small. It leads to extremely tight requirements for the parameters of the measuring system [18,19].

The lack of attachment of the electrode to the body of the monitored person leads to increased susceptibility of the measuring system to so-called moving artifacts [18]. In general, the electrode may move relative to the body surface in any and complex way. The design of the capacitive electrode has been studied for many years but is still challenging and under research [20,21].

cECG measurements using a system embedded into an armchair are considered in the paper. This application seems to be a relatively widespread one [18]. Aside from the measurement properties of the capacitive electrode, the lead configuration used is also important. When considering the application of the method with systems built into furniture, e.g., an armchair or a chair, the most commonly used is a lead analogous to the first Eindhoven one, but with electrodes placed in the area of the back and at different heights below the shoulders [22,23]. On the other hand, the ECG lead referred to as monitoring is widely used in many applications. However, it is selected individually for each examined person [20,24,25]. This approach is validated for capacitive measurements and armchair-embedded systems more rigorously in this paper.

The paper is organized as follows. Section 2 describes the numerical model for studying the potential distribution on the surface of the chest (the so-called simple ECG problem), a model of the capacitive electrode that makes it possible to relate the value of the electrode potential with the potential distribution on the body surface and the impact of its relative movement on generating motion artifacts, a description of the built measurement system and measurement procedures, and the group of volunteers participating in the research. Section 3 presents exemplary potential distributions on the thorax and electric field intensities that determine the optimal lead configuration. Section 4 contains an extensive discussion of the results obtained and the conclusions drawn from these results. The article ends with the statement that a lead appropriately selected for the examined person enables the use of cECG, in the application of the built-in system, to monitor people in conditions of moderate mobility while sitting in an armchair.

## 2. Materials and Methods

Numerical and experimental studies were carried out to select the best leads for a person sitting in an armchair. To obtain the optimal placement of the electrodes on the back of the monitored person, which was equivalent to the highest R-wave value measured for a given lead, simulation tests were carried out using the finite element method (FEM) [26]. To verify the results, a measurement system allowing simultaneous recording of cECG and ECG was developed. Finally, a group of 14 volunteers was examined to prove the feasibility of the concept.

### 2.1. Potential Distribution on the Thoracic Surface

Optimal localization of the electrodes should follow from the potential distribution on the body of the examined person. In turn, the potential distribution is determined by material (electrical parameters, i.e., electric conductivity, electric permittivity) and geometric factors. This problem is known in the literature as a forward problem in electrocardiography. The analysis of ECG forward problems can be found in [27] while general considerations are described in [28]. According to the above-mentioned papers, the electric potential distribution, φ(x,y,z), shortly assigned as φ in the thorax, can be calculated using the partial differential equation [22,29]:(1)$$\nabla \cdot \left(\sigma \nabla \varphi \right)=I_{SV}$$
where:(2)$$I_{SV}=-\nabla \cdot J_{h}$$
and $J_{h}$ denotes the cardiac source (so-called active currents). The current density that is normal to the thorax surface, S, also has to fulfill a boundary condition:(3)$$\frac{\partial \varphi }{\partial n}=0_{onS}$$
where n is a unit vector normal to S.

The above problem was solved using the FEM. The model of the thorax was built based on CT scans from a 68-year-old man. The scans covered a chest area of 42.3 cm height, the distance between successive scans was 0.5 mm, and the resolution of each scan was 512 × 512 (px). The first step was the segmentation of individual images for scans. By selecting the appropriate range of Hounsfield units (HU), i.e., defining the histogram window corresponding to a given tissue, individual internal organs were selected. For each scan, a set of masks corresponding to the individual organs was created. For example, the range of values selected for a lung mask was 1024–500 HU. Segmentation based on Hounsfield scale values were only the first approximations of the designed masks. In general, tissue differentiation within one organ was quite significant. Moreover, the issue was complicated by the network of blood vessels encircling the internal organs and, in the case of the lungs, the bronchial tree and bronchioles. The resulting masks were, therefore, not smooth, contained many “holes”, and covered areas that did not belong to the organs concerned. It was, therefore, necessary to further process them. Various methods of image processing have been used for this purpose, e.g., morphological operations, such as erosion, dilation, opening or closing, logical operations, and others [30]. Based on the masks obtained during the segmentation, a three-dimensional model of the chest was generated. It contained the heart, aorta, major blood vessels, right lung, left lung, liver, spine, and ribs. To reduce the number of elements in the FEM model, the obtained model had to be simplified. It was decided to remove the smaller blood vessels and ribs, leaving only the largest internal organs with their surfaces smoothed.

### 2.2. Capacitive Electrode

An ideal measurement system should have used electrodes with infinitesimally small areas (points). However, such electrodes are characterized by very high impedances. This problem has been solved for contact electrodes by applying appropriate technological processes that reduce and stabilize the electrode impedance value. When considering non-contact measurements this approach is not possible. As a result, an acceptable signal quality is achieved by increasing the input impedance of the measurement system. For the problem to be technically solvable, electrodes with a diameter of 2 cm are accepted. The coupling capacitance of such an electrode, at the first approximation, is described by Equation (Equation 4):(4)$$C=\varepsilon_{0}\varepsilon_{r}\frac{S}{d}$$
where *S* is the electrode area, *d* is the distance between the electrode and the subject’s body, and $\varepsilon_{0}$, and $\varepsilon_{r}$ is the electrical permittivity of the free space and relative permittivity of the material inserted between the electrode and the body, respectively.

Even though there is the distribution of the potential under the electrode on the surface of the monitored person’s body, the electrode itself is equipotential. This means that its potential is described by one value, $V_{e}$. The value of $V_{e}$ depends both on the parameters of the measurement system, e.g., its input impedance, and on the shape and distance of the electrode from the body, as well as on the electrical properties of the materials between them. To obtain an estimate, ignoring the influence of the measurement system, it can be approximated by Equation (Equation 5) [31,32]:(5)$$\int \int_{S_{e}}\left[V\left(x,y,z\right)-V_{e}\right]g\left(x,y,z\right)dS=0$$
where V(x,y,z) is the potential distribution at the body’s surface located beneath the electrode, g(x,y,z) is the specific admittance (admittivity) distribution between the electrode and the body, and $S_{e}$ is the electrode area. In the considered case, it reduces to (Equation 6)
(6)$$g\left(x,y,z\right)dS=j\omega C_{e}\left(x,y,z\right)$$
where $C_{e}\left(x,y,z\right)$ is the coupling-specific capacitance distribution, ω is the frequency, and g(x,y,z) is the admittivity.

As a result, the potential of the electrode is described by Equation (Equation 7):(7)$$V_{e}=\frac{\int \int_{S_{e}}V\left(x,y,z\right)C_{e}\left(x,y,z\right)dS}{\int \int_{S_{e}}C_{e}\left(x,y,z\right)dS}$$

Equation (Equation 7) may be helpful in estimating the sensitivity of the $V_{e}$ potential to the electrode movement, i.e., in the assessment of motion artifacts. The electrode sensitivity to motion artifacts can be defined as the directional derivative of $V_{e}$.

### 2.3. Measurement System

The measurement system consisted of two channels and allowed simultaneous recording of cECG and ECG signals (Figure 1). Both measurement channels were electrically isolated from the Analog Discovery 2 system. It reduced the possible interaction between them. Such a solution made it possible to compare the cECG signal quality to the simultaneously recorded ECG.

It should be noted that the term capacitive electrode is understood as meaning both the electrode itself and the voltage follower, characterized by a very high input impedance, connected directly to it. For the classical approach, an Ag/AgCl electrode is meant.

### 2.4. Experimental Studies

A two-stage experiment was carried out. The first was the simultaneous recording of cECG and ECG signals for certain (previously selected) leads. They were selected based on the determination of the electrical axis for each person when sitting in the armchair. This was achieved with a combined electrode system. The non-contact and contact electrodes were placed side-by-side and both were attached to a belt worn over the test person. The non-contact electrode was covered with 1-mm-thick cotton to simulate real conditions of non-contact measurements. Then, the quality of the data obtained were evaluated by means of SNR. The SNR was defined as $20\cdot log(V_{R}/\sigma_{N})$ where $V_{R}$ stands for the average value of the R-wave amplitude and $\sigma_{N}$ for a standard deviation of the noise. Th noise was measured between the T and P waves of the recorded cECG and ECG. The relation between the mean values of the R wave amplitudes as measured on the chest and back was also determined. The last experiment in this stage of the research consisted in determining the dependence of changes in the R-wave amplitude in response to small changes in the direction of the lead, by about 15 and 30 degrees in relation to the optimal one. The second experiment was performed with the use of capacitive electrodes attached to the backrest of the armchair. The electrodes creating the optimal lead were determined earlier, before the examination, for each examined person. The examined person, wearing a cotton T-shirt, was asked to sit in a chair and watch a movie for 15 minutes. A moderate body movement, preserving at least a slight contact with the electrodes, was suggested. These long-term cECG signals were used to assess the quality of the detection of the QRS complexes. Then, a software detector based on the Pan–Tompkins approach was used to detect the QRS complexes [33,34,35]. The quality of the detection was described with appropriate statistical parameters [36,37].

The same group of 14 people participated in both studies. The group consisted of 12 men and 2 women. The mean age was 35 and the mean body mass index was 29. All subjects received detailed information about the study objectives and any potential adverse reactions, and they provided written informed consent to participate in the study. The experimental protocol and the study were approved by the Bioethics Committee at the Regional Medical Chamber in Gdańsk.

## 3. Results

The potential distribution associated with electrical heart activity was modeled using an equivalent dipole approach [22]. The value of the dipole moment was $2.0\times 10^{-5}$ A · m. This value was in agreement with the data available elsewhere, e.g., [38]. The initial, reference values of the azimuth and elevation angles were 5/6π and 3/4π, respectively. Please note that they refer to the coordination system used in the developed thorax FEM model (Figure 2). It contained a heart filled with blood, lungs, and a thoracic wall, each with different (yet appropriate) conductivities. The remaining organs located below the diaphragm were included as electrically uniform tissues.

The model was developed based on images obtained by means of computer tomography (CT) measurements. The CT data were segmented and exported to *.stl files using the Slicer software. To reduce the mesh complexity, quadratic edge collapse decimation was performed, together with Laplacian smoothing. Both operations were performed using the MeshLab software. The final model contained over 110,000 tetrahedral elements. The x-coordinate was oriented from the left to right shoulder (frontal plane), the y-coordinate from back to front (sagittal plane) and the z-coordinate from bottom to top. The center of the heart base, i.e., (0.1, 0.03, 0), was set as the position corresponding to a mid-respiration state.

The conductivity value assigned to each tissue included in the model was selected according to the frequency spectrum of the ECG. Thus, the conductivity of the heart muscle, the thoracic wall, the blood, and the abdominal part of the body (the part below the diaphragm) were, respectively, equal to 0.2 S/m, 0.1 S/m, 0.5 S/m, and 0.1 S/m [22,39]. The lungs’ conductivity was fixed and equal to 0.05 S/m. The respiration effect on the electrical conductivity of the lungs was omitted.

The potential distribution on the chest and the back for different parameters of electrical heart axis exhibit repeatable (however, changeable) patterns (Figure 3). Based on the potential and electric field distributions on the thoracic surface and the electrode locations, it is possible to estimate the lead voltage and its sensitivity to the electrode movement, i.e., motion artifacts.

The sensitivity of the electrode potential, $V_{e}$, to a change in its position (moving artifacts) depends both on the potential and specific capacitance distributions and their derivatives. The first case is that the electrode simply moves away from the body surface. Then, in Equation (Equation 7), the change of $C_{e}\left(x,y,z\right)$ should be assumed while the potential distribution is fixed. The value of this capacity is decreasing with the increased distance of the electrode. In the first approximation, it can be assumed that the potential distribution on the surface of the body, which determines the electrode potential together with the capacitance, does not change. In the second case, the electrode moves along the body while maintaining a constant distance from it. In this case, the specific capacitance can be assumed to be constant. As a result, the electrode potential depends on the directional derivative of the body surface potential, i.e., on the components of the electric field intensity (Figure 4). Thus, as the first approximation, the electric field components $E_{x}$, $E_{y}$, and $E_{z}$ could be utilized to estimate the directional derivative. The regions with high values, as opposed to regions with low electric field values, are more sensitive to motion artifacts.

The results of the numerical studies made it possible to select the configurations of the leads used in the experimental studies. For the potential distribution presented in Figure 3c, the optimal lead is presented in Figure 5a. The next two lead configurations, Figure 5b,c, were selected to evaluate the dependence of the lead voltage on its relation to the electrical heart axis. In turn, the optimal lead presented in Figure 5d was obtained for the person characterized by a more horizontal heart axis. The associated suboptimal leads are not presented. Other lead configurations were also tested to verify the correctness of the assumptions and the theoretical results obtained.

An electronic circuit directly attached to the capacitive electrode consisted of two operational amplifiers; however, one IC was utilized (LMP 7702) (Figure 6a). A very high input impedance was achieved by using a bootstrap voltage follower. The other operational amplifier was utilized to create a shielding signal (active shielding approach).

The capacitive electrode was developed based on a four-layer PCB. The measuring–internal part was surrounded by a shielding ring (Figure 6b). which, together with one of the middle layers of the PCB, was connected to a shielding signal.

The frequency characteristics of the amplifier were determined by the AD 8232 integrated circuit. The measured signals were digitized at a frequency of 2.3 kHz. Then, to reduce the amount of data, they were downsampled using an averaging filter to 72 Hz. Based on the data recorded during the first experiment, the signal quality was determined by the signal-to-noise ratio, SNR. The SNR value was estimated for each measured lead (Figure 7). The SNR values (mean ± SD) for the optimal, non-contact, and contact configurations were 19.4 ± 1.85 and 35.29 ± 1.98 dB, respectively. In the case of the worst configuration, it was not possible to detect the QRS complexes reliably for the non-contact measurement. The relation between the R-wave amplitude as measured on the chest and the back was also determined for the optimal leads (Figure 8).

The distribution of the R-wave amplitude, as recorded by different leads on the back for the whole examined group, showed similar variability (Figure 9).

The long-term cECG recorded using the optimal lead was characterized by an acceptable level of noise and artifacts (Figure 10).

The recorded cECG signal was contaminated by noise and motion artifacts; however its quality was acceptable and enabled further processing, e.g., QRS detection (Figure 11).

The number of detected QRS complexes was slightly higher than the real number(Table 1), which was estimated manually by an experienced technician.

## 4. Discussion

Capacitive electrocardiography is still under development and appears to be a promising technique. Research is being conducted in several main directions. One is to reduce the level of distortions and noise to make it possible to interpret other ECG waves apart from the QRS complex. The research presented in the paper partly fits in this direction. We proposed to personalize the measurement procedure by selecting the optimal lead and minimizing motion artifacts simultaneously. The optimal electrocardiographic lead is understood as the one that results in the recorded ECG signal having the best relation of the R-wave amplitude to the noise standard deviation value. To determine its configuration the potential distribution on the thorax was calculated by the FEM method. The potential distribution on the body surface developed by the electrical activity of the heart depends on the point that is measured with respect to the phase of the depolarization wave propagation through the ventricles. In the literature on the forward problem, this field is considered to be largely dipole in nature. Therefore, the adoption of such a signal source model can be considered justified. The calculation costs are significantly reduced while the obtained results are sufficient to assess the basic properties of the potential distribution. The direction of the electrical heart axis differs between individuals and both the elevation and the azimuthal angles may have different values. This influences the shape of the potential distribution on the examined person’s back (Figure 3). When the heart vector is directed more forward to the chest wall then the potential distribution on the back is more homogeneous and simultaneously negative. As a result, the achievable lead voltage may be significantly lower than in other cases. As the azimuth angle increases, the potential distribution on the back becomes more diverse and bipolar. The elevation component has a similar effect on the potential distribution. Thus, the thoracic potential distribution, which in general depends on many other factors not discussed here, is unique to each examined person. The electric field distribution has the similar properties (Figure 4).

Based on the knowledge of the distribution of both the potential and the electric field, and taking into account Equation (Equation 7), several important conclusions can be drawn regarding the properties of capacitive measurements.

The condition described by Equation (Equation 5) is equivalent to the assumption that the total current flowing between the electrode and the body is zero and that application of the electrode does not significantly change, the potential distribution in the body. It is fulfilled when the specific admittance between the electrode and the body is significantly lower than the specific admittance of the body. It is equivalent to the statement that the current density flowing from the body to the electrode and then in the opposite direction is much lower than the local current density in the adjacent part of the body. Thus, taking into account that the resulting admittivity (Equation (Equation 6)) is much lower than that of the tissues’ involved, the proposed model could be considered to be appropriate.

The analysis of Equation (Equation 7), linking the value of the electrode potential, $V_{e}$, with the specific capacitance, $C_{e}\left(r\right)$, and the surface distribution of the subject’s body potential V(r), reveals that the surface distribution of the electric field is also important when considering the sensitivity of the electrode to motion artifacts. It should be underlined that Equation (Equation 7) is acceptable when the electrode diameter is larger than its distance from the subject’s body surface. Moreover, the sensitivity of the electrode potential to its displacement is rather complex as it requires the determination of the directional derivative of Equation (Equation 7), i.e., $dV_{e}/dr$. Generally, it is equivalent to the requirement that the Leibniz integral rule is fulfilled. However, it is possible to draw conclusions when considering the simplified cases of the electrode’s displacement while assuming that the limits of integration for these cases do not depend on the displacement. Firstly, the movement of the electrode to or from the object can be assumed to take place under the conditions of an unchanging potential distribution V(r). Secondly, it can be assumed that the movement of the electrode takes place along the surface of the examined subject. In the first case, the change in capacitance can be estimated using Equation (Equation 4). In the latter case, it can be assumed that the capacitance $C_{e}\left(r\right)$ is invariant, and as a result, $dV_{e}/dr$ depends on the tangent component of the electric field on the chest surface. In general, the reduction of the value of $dV_{e}/dr$ can be achieved by reducing the value of the electric field (Figure 4). Thus, both conditions of optimal electrode placements are met when electrodes are placed as far as possible from each other and in regions of potential opposite signs and simultaneously characterized by a low tangential electric field.

The electrode’s dimensions are also important when assessing both its potential and the susceptibility to the generation of motion artifacts. As it results from the model adopted, its potential may have different values. When a uniform coupling is met (a uniform specific capacitance), it can be deduced from Equation (Equation 5) that the electrode potential is equal to the spatial average of the potential distribution. Moreover, when the subject’s surface is equipotential, e.g., it is equal to *V*, the potential of the electrode is also equal to *V* which also directly results from Equation (Equation 5). In the opposite case, i.e., inhomogeneous potential distribution, each movement of the electrode is associated with a change in its potential. This change also depends on the nature of the electrode movement, the potential distribution on the surface of the body, the electric field associated and the electrode dimensions. This phenomenon could be reduced by minimizing the electrode size. However, the size of the electrode is determined by parameters of the measurement system, e.g., input impedance.

A close inspection of the potential and electric filed distributions enabled determining the optimal lead, i.e., the electrode localization, for the measurement performed on the back of the examined person. The optimal lead can vary greatly from person to person (Figure 5a,d). Thus, the lead that is optimal for one person may be suboptimal for another. As a result, for a person with an electrical heart axis parallel to the Eindhoven II lead, the appropriate lead was selected (Figure 5a). This lead was supplemented by two others to validate the accepted assumptions (Figure 5b,c). The optimal lead direction (Figure 5d) for another person was between the first and second Eindhoven leads.

To perform measurements, the system containing two, electrically isolated channels was developed (Figure 6). The AD 8232 integrated circuit was utilized in both channels. To avoid interference between the channels, galvanic isolation was used. The measuring channels differed in the type of electrodes. One was equipped with capacitive electrodes and the other with classic Ag/AgCl ones. The capacitive electrodes were constructed based on a bootstrap circuit (Figure 6a,b) characterized by a very high input impedance. In turn, the frequency characteristic was determined by AD8232 properties (Figure 6c). It should be noted that in the literature one can find systems characterized by a higher input impedance and, at the same time, a lower noise. Nevertheless, the parameters of the custom-made electrodes were sufficient to carry out the intended experiments.

The developed measurement system was used for the experimental measurements (Figure 6). It consisted of two measurement channels, each electrically separated from the other, for measuring cECG and ECG simultaneously. The developed capacitive electrode buffer of high input impedance was based on a so-called bootstrap circuit. This part of the electronic measurement circuit was directly attached to the capacitive electrode. However, the achieved parameters can be improved when compared to other solutions. Nevertheless, the quality of the recorded signals obtained was sufficient to perform examinations (Figure 7). However, the noise level was the lowest for the optimal lead (Figure 7a) and much higher for the sub-optimal ones.

A relation between the amplitude of the R-wave as measured at the chest and the back shows that the former is larger (Figure 8). This was in agreement with simulation studies. This is probably due to the position and alignment of the heart in the thorax. However, we did not explore this issue. Mostly young people with low BMIs participated in the study. The examination of people characterized by higher BMI values would probably change this relation. The distribution of the relations between the R-wave amplitude as measured by means of the optimal lead and two others for the examined persons are shown in Figure 9. There is a natural interpersonal variability of the R-wave amplitude when considering the optimal lead. The distribution between R-wave amplitudes, as obtained using different leads, also varies significantly for almost all examined subjects, even when the changes in the lead direction were not significant. However, it should be underlined that the estimation of the angles was not precise in our study. Nevertheless, the data obtained exhibit a clear dependence of the R-wave amplitude on the utilized lead direction in relation to the actual electrical heart axis. This statement is obvious and in agreement with the theory of electrocardiography. Finally, during the second stage of the study, each person was asked to sit in the armchair containing embedded electrodes whose positions were matched individually based on earlier measurements. The person was resting for 5 minutes before the cECG recording started for 15 minutes (Figure 10a). The examined person was watching a movie and moderate body movement was allowed, which was easily visible in the recorded signal (Figure 10b). The obtained signal was processed and QRS detected (Figure 11). The statistics were calculated (Table 1). It followed from the study that the quality of the signals made it possible to further process them, which provided information on the heart rate. However, this result is to some extent optimistic because the examinations were supervised, thus the conditions were not totally natural.

To reference the results presented in the paper to the achievements of other scientific teams, a short collection is included in the Appendix A (Table A1).

## 5. Conclusions

An appropriate selection of the electrocardiographic lead makes it possible to improve the quality of the recorded cECG signal as evaluated by the R-wave amplitude in relation to the noise of the measurement system. Localization of the electrodes in the region of the low tangent electric field developed on the subject’s body surface may also reduce motion artifacts. Further improvement can be achieved by minimizing the electrode dimensions; however, this approach is conditioned by the increased demands on the performance of the measurement system. The developed measurement system embedded in the armchair’s backrest provided long-term monitoring of the heart rate in people making moderate movements. The QRS detection rate achieved in the study is satisfactory. Nevertheless, there is still room for improvement and a more robust QRS detection algorithm may reduce the number of false positive detections.

## Figures and Tables

**Figure 1 sensors-22-09475-f001:**
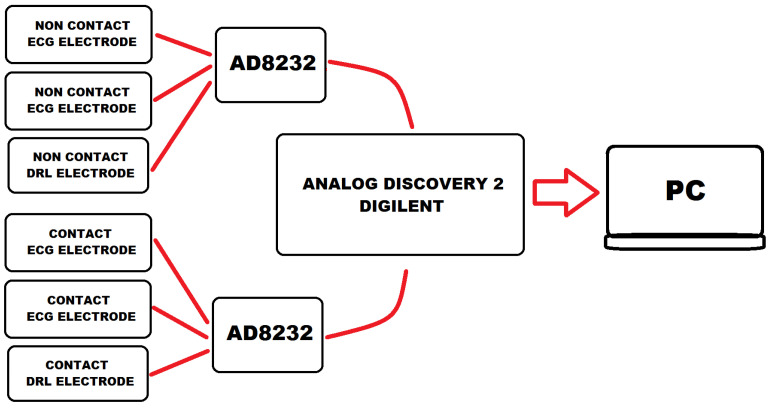
Measurement system enabling recording contact and non-contact electrocardiograms.

**Figure 2 sensors-22-09475-f002:**
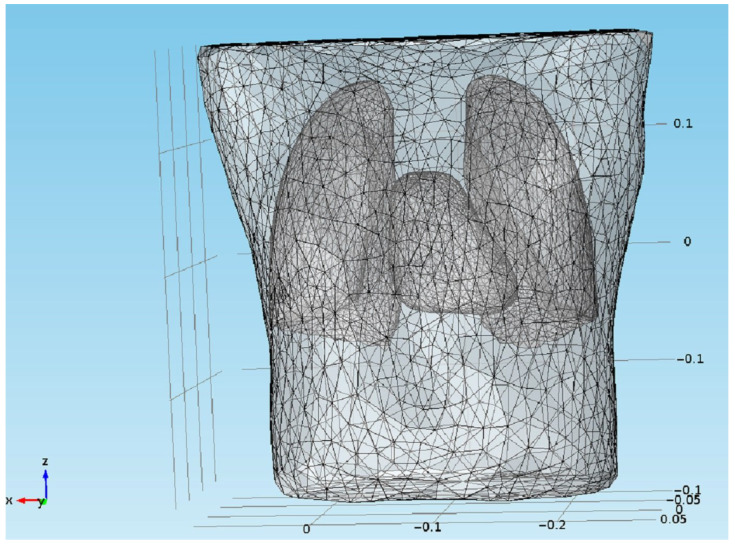
Developed FEM model used in the study.

**Figure 3 sensors-22-09475-f003:**
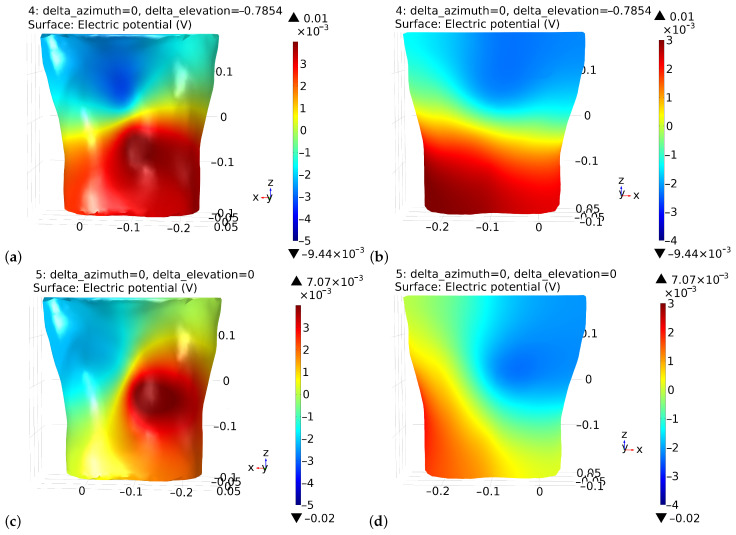
Exemplary results of the potential distribution, on the chest and back, respectively, at the moment of the R wave of the ECG for two selected heart axis directions which differ in the azimuthal angle value, (**a**,**b**) present the potential distribution, respectively, on the chest and the back for azimuthal angle π/2, while (**c**,**d**) present the potential distribution, respectively, on the chest and the back for azimuthal angle 3π/4. Note that the azimuthal angle is determined in reference to the x-axis and that different scales are accepted for the figures.

**Figure 4 sensors-22-09475-f004:**
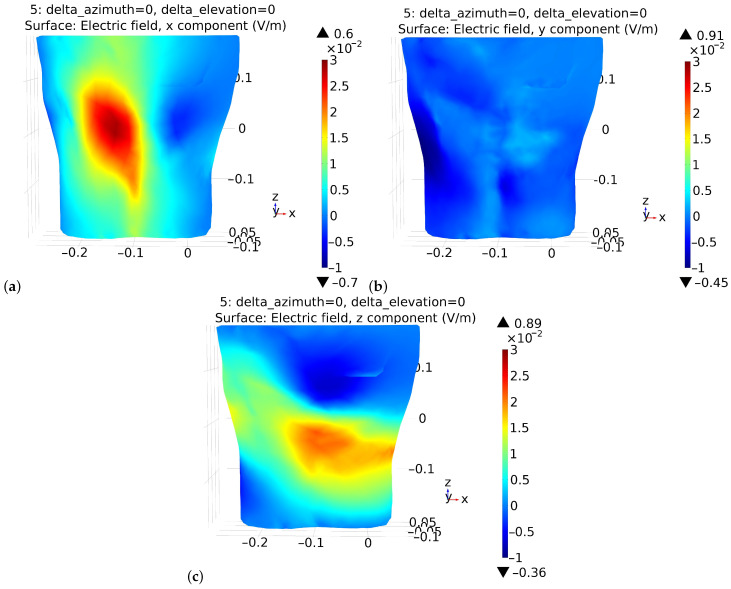
Electric field distribution on the back (**a**) Ex, (**b**) Ey, and (**c**) Ez components, respectively.

**Figure 5 sensors-22-09475-f005:**
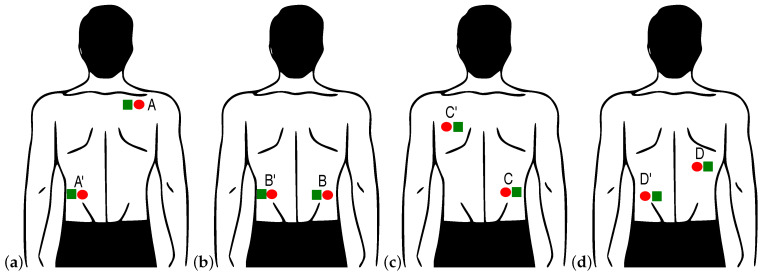
(**a**–**d**) Configuration of the leads (see text for details) used in the study for measuring ECG on the back (illustrative figures). Two signals were recorded simultaneously using contact (red circle) and non-contact (green square), respectively. Reference to sub-figures inside text.

**Figure 6 sensors-22-09475-f006:**
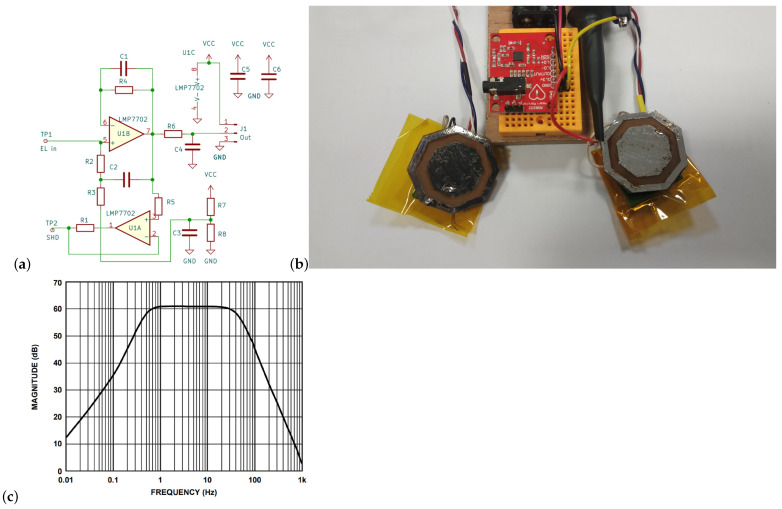
Developed capacitive electrode: (**a**) circuit diagram, (**b**) measurement system (DRL electrode not shown), (**c**) frequency characteristic of the measurement system as determined by an AD8232.

**Figure 7 sensors-22-09475-f007:**
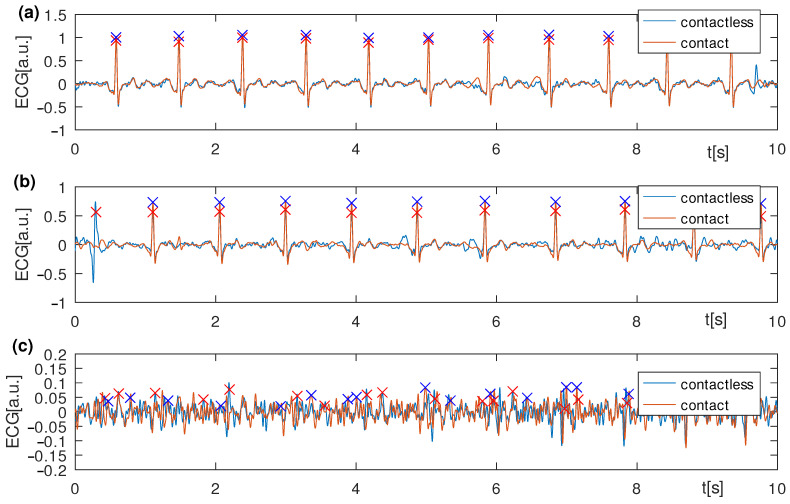
Simultaneous recording of ECG and cECG using different leads presented in Figure 5; (**a**) A’A lead, (**b**) B’B lead, and (**c**) DD’ lead. The signals were recorded for the person; the heart’s electrical axis was almost parallel to the I Eindhoven lead.

**Figure 8 sensors-22-09475-f008:**
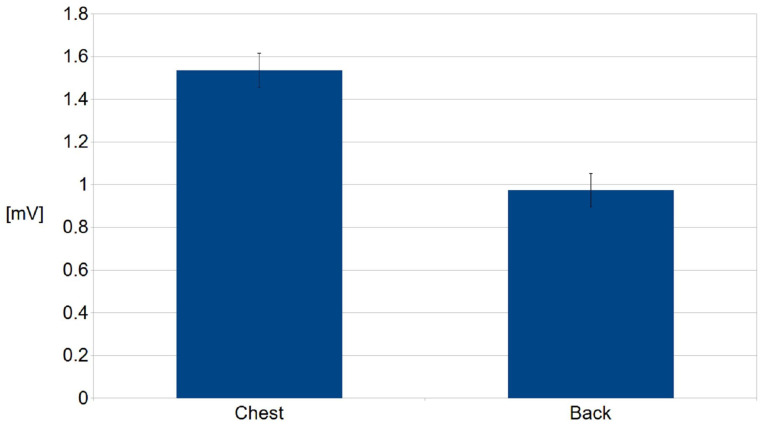
Relation between averaged values of R-wave as measured on the chest and the back using the optimal leads for the whole group of volunteers.

**Figure 9 sensors-22-09475-f009:**
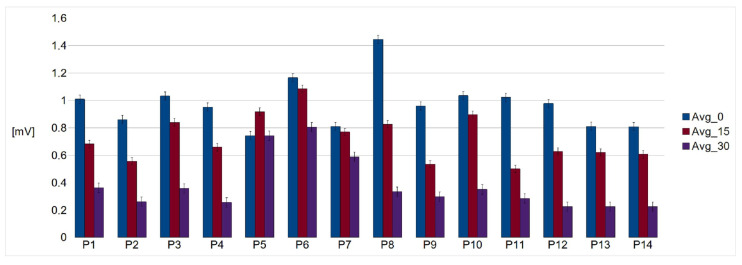
Relation between the R-wave amplitudes as measured for different leads including the optimal one (marked blue) and two others. $Avg_{0}$ represents the optimal lead, while $Avg_{1}5$ and $Avg_{3}0$ represent the leads, forming angles of 15 and 30 degrees from the optimal ones, respectively.

**Figure 10 sensors-22-09475-f010:**
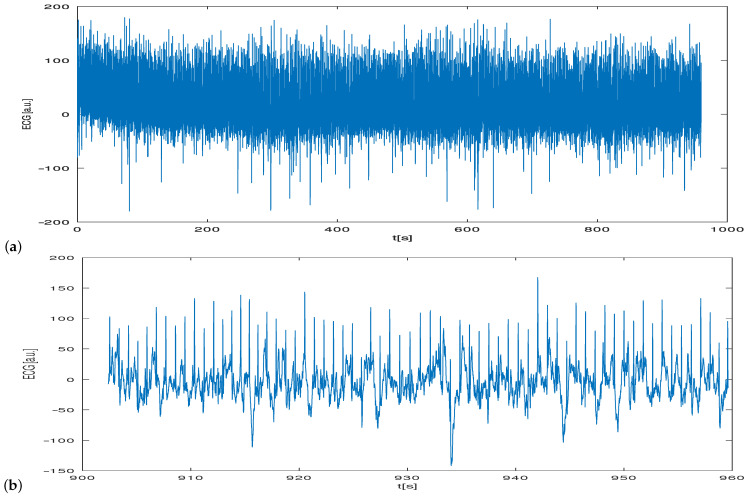
Example of cECG signal recorded for a period of 15 minutes (**a**) and part of the signal shown for a different time scale (**b**).

**Figure 11 sensors-22-09475-f011:**
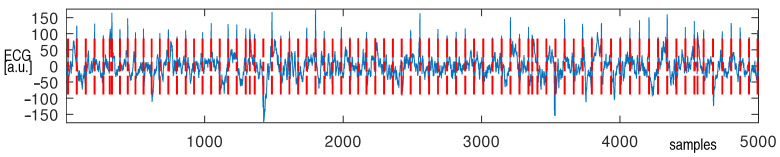
Detection of the R wave in the QRS complex—vertical red lines indicate the positions of the R-wave detected.

**Table 1 sensors-22-09475-t001:** Combined overall results of QRS-complex detection.

Actual No. QRS	TP	FN	FP	TE	SE	PPV
14,087	13,948	139	259	398	99.01%	97.22%

TP—true positive, FN—false negative, FP—false positive, TE—total error, SE—sensitivity, PPV—prognostic positive value.

## Data Availability

Not applicable.

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
