# Peer review of "Non-Contact Monitoring of ECG in the Home Environment—Selecting Optimal Electrode Configuration"

_sensors, 2022, doi:10.3390/s22239475_

Round 1
Reviewer 1 Report
In this manuscript, the authors investigated optimal electrode configuration for ECG measurement using non-contact electrodes embedded in an armchair. They analyzed the potential distribution on thoracic surface to optimize the location of the electrodes. Here are some comments
- All figures are in low quality and they should be improved. The text is blurry and it is very hard to follow and understand.
- The driven-right-leg electrode was not mentioned. How does that effect the differential potential of two other electrodes?
- The authors did not conclude about the best electrode placement for getting high SNR and the statement “electrode placement depends on each examined person” is vague with no evidence provided.
- It’s hard for audience to follow this manuscript. Please revise and make it more organized, especially the Materials and Methods section.
Minors
- In the abstract, please describe how SNR and other parameters were improved quantitatively with the optimal number of leads.
- All formulas should be rewritten with a proper math format.
Author Response
Dear Madame/Sir,
please find attached the response to you comments and suggestions.
Kind reagrds
Jerzy Wtorek

Reviewer 2 Report
The authors present the article entitled “Noncontact monitoring of ECG at home environment selecting optimal electrode configuration”
A non-contact paper electrocardiography (cECEG) using a system embedded in a chair is considered in the paper.
The article presents the following concerns:
-
I suggest restructuring the Abstract. It is ambiguous, and the methodology is not described. I also recommend adding quantitative values to highlight the findings of the work. Lines 9-13 are the same information as lines 25-30.
-
Improve the keywords.
-
Avoid using apostrophes.
-
Introduction section: The objective of the manuscript is not clear. I suggest improving it by highlighting the novelty and contributions of the work.
-
Equations and variables are hard to follow. For example, in line 85, “Jh”, the “h” must be a subscript. In Equation 3 there is a missing space, I_SV and “rho” are not declared. Check all the equations are correct and all the variables are declared.
-
Lines 91-113: This may be part of the Results section. In Section 2, it is recommended to describe theoretically how the CT obtains the model.
-
Equation 7: You call this Equation a relationship in all text, making it difficult to follow. Please refer as an Equation.
-
Line 26, which is about ECG examples, could be justified with this reference regarding up-to-date ECG monitoring: Speed controller-based fuzzy logic for a biosignal-feedbacked cycloergometer.
-
Line 69 could describe the FEM topic by giving certain examples as Finite element method and cut bar method-based comparison under 150ffi, 175ffi, and 310 ffiC for an aluminium bar; Finite-element simulation for thermal modeling of a cell in an adiabatic calorimeter; Form-finding analysis of a class 2 tensegrity robot.
-
All Figures and tables must be mentioned in the main text.
-
What is the purpose of Figure 10? Its description in lines 279-280 is poor.
-
In the discussion section, please add a table that compares the findings of the work vs. the already reported state of the art.
-
Figure 11: axis titles are missing.
-
It is recommended to describe the structure of the text at the end of the introduction.
-
Add hyperlinks to tables, figures, and references.
-
Make the subtitle of line 171 in italics like the previous ones
-
Figures 3, 4, 7, and 10 must be vectorized to see the details.
-
Please read the guide for authors to check the format of tables and content of the manuscript
The following misspelling should be checked:
-
line 25: It appears that you are missing a comma with the interrupter “however”. Consider adding the comma: “.However, the….”
-
line 39: “with the use of…” should be rewritten as “using…”
-
line 49: “area of the electrode…” should be rewritten as “electrode area…”
-
line 54: The phrase “in relation to” may be wordy. Consider changing by “about”, “to”, “with” or “concerning”
-
line 95: It appears that the article “the” is unnecessary before “figure 1” in this sentence. Consider removing the article.
-
line 138: “in order for…” should be rewritten as “for…”
-
line 220: “on the basis of…” should be rewritten as “based on…”
-
line 329: Apostrophes must be avoided, for example “subject’s”
-
line 346: “in reference to…” should be rewritten as “about…”
-
line 378: “be helpful in determining…” should be rewritten as “help determine…”
Author Response
Dear Madame/Sir,
please find attached the response to your comments and suggestions.
Kind regards
Jerzy Wtorek

Reviewer 3 Report
This manuscript applies computational method to find the optimal electrode of noncontact ECG configuration. The method is based on FEM analysis. After numerical simulation, real-data analysis is performed on different subjects. Results are discussed in detail.
The most cherished merit of this draft is the detailed explanation on the mathematical derivations. The underlying application of this draft is novel, too.
However, this draft also has several drawbacks, including text writing, structure organization, and lack of a few key points in results/discussion.
Text and Writing:
1. Please pay attention to some of the spellings. For example (not limited to):
In the Abstract, “....of measurement system. E.g., to make the...” Be cautious! The phrase “e.g.,” cannot be used at the beginning of one sentence. A possible replacement is “For example”. Also, in Section 2 above Equation (5), “It means that its potential is described by one value, e.g. Ve.” Here should be “e.g.,”, with a comma.
In Section 2, after Equation (2), ‘and Jh denotes the cardiac source...’ Be careful! In the text, Jh should be put in mathematical form, i.e., h is in the subscript. Another similar one is under Equation (3): “.....was equal to 2.0*10-5Am”. Does this “10-5” refer to the -5 power of 10? If yes, please clarify it. Please follow scientific writing style, check ALL other places in the text, and re-write the in-text mathematical expression in standard form.
2. Please improve the size and solution of Figures 7 and 10, to make them clear.
3. Please reform the layout of Table 1. There is no upper-line and bottom-line of it, which makes it hard to differentiate from main text. A possible example is the IEEE writing style manual (reference only):
M. Shell, How to Use the IEEE LaTex Class, Jpournal of Latex Class Files, Vol. 14, No. 8, August 2015. Section X.c
Paper Organization:
The Section 2 and 3 needs to be re-organized for improved readability and clarity
1. In Section 2, please change the title of “Experimental studies”, as it is not yet related to final experiments. A possible alternative is “Subject Information for Study”.
2. The Section 3 and 4 can be incorporated together. Also, the content of discussion is too long and a bit verbose, not matching the presented experimental results.
3. Following above-mentioned points, the new Section 3 Results can be organized like this (reference only): 1) numerical simulation; 2) subject studies; 3) discussion.
4. Also, please summarize the simulation parameters together before showing the simulation outcomes, for better readability
5. This paper helps explain explanation on “Each mentioned application demand a specific construction of measurement system [added reference]
G.Peng, M. Nourani, J. Harvey and H. Dave., “Personalized feature selection for wearable EEG monitoring platform”, 20th IEEE international conference on bioinformatics and bioengineering (BIBE), October 2020. pp. 380-386
Experimental Results
A major concerns is the lack of demonstration on real-subject results. Especially, when authors claim the subject-dependent electrode configuration, how to show this point? A possible way is that, using different subjects, and show their corresponding electrode selection results.
Author Response

(The authors gave the same response as above.)

Round 2
Reviewer 2 Report
The authors have addressed my comments
Reviewer 3 Report
looks good for this version